# Novel Cross-Cancer Hub Genes in Doxorubicin Resistance Identified by Transcriptional Mapping

**DOI:** 10.3390/biomedicines13102527

**Published:** 2025-10-16

**Authors:** Arseny D. Moralev, Oleg V. Markov, Marina A. Zenkova, Andrey V. Markov

**Affiliations:** Institute of Chemical Biology and Fundamental Medicine, Siberian Branch of the Russian Academy of Sciences, 630090 Novosibirsk, Russia; arseniimoralev@gmail.com (A.D.M.); markov_oleg@list.ru (O.V.M.); marzen@niboch.nsc.ru (M.A.Z.)

**Keywords:** chemoresistance, MDR, doxorubicin, cancer, hub gene, regulome, transcriptomics, gene network

## Abstract

**Background:** Doxorubicin (DOX) is a widely used chemotherapeutic agent, but its efficacy is often limited by cancer cell resistance. Although multiple DOX resistance mechanisms have been characterized, the global transcriptomic alterations underlying this phenomenon remain poorly understood. The aim of this work was to determine whether a common transcriptional response associated with DOX desensitization exists across tumor cells of different origins and to identify the core elements of this response. **Methods:** We performed an integrated bioinformatics analysis, including: analysis of independent transcriptomic datasets (comparing DOX-resistant neuroblastoma, breast, and cervical carcinoma cells to their DOX-sensitive counterparts), functional annotation of differentially expressed genes, reconstruction and topology analysis of gene networks, text mining, and survival analysis. The findings were validated through in vitro functional tests, RT-PCR, and analysis of the Cancer Therapeutics Response Portal and The Cancer Genome Atlas. **Results:** We showed that DOX resistance in cancer cells is associated with cytoskeletal reorganization, modulation of cell adhesion, cholesterol biosynthesis, and dysregulation of mTORC1, Wnt, and Gβγ signaling pathways. Network analysis identified a conserved regulome of 37 resistance-linked genes, with *GJA1*, *SEH1L*, *TCF3*, *TUBA4A*, and *ZYX* emerging as central hubs (mean degree: 8.7–19.7; mean fold change: 2.4–21.3). Experimental validation in DOX-resistant KB-8-5 cervical carcinoma cells and their sensitive counterparts (KB-3-1) confirmed enhanced cellular adhesion and reduced intracellular cholesterol levels associated with chemoresistance, supporting our in silico findings. A detailed follow-up analysis verified the upregulation of these hub genes in chemoresistant cells and their correlation with poor clinical outcomes across multiple cancer types. **Conclusions:** This integrative analysis identifies conserved transcriptomic signatures of DOX resistance and highlights hub genes *GJA1*, *SEH1L*, *TCF3*, *TUBA4A*, and *ZYX* with potential as predictive biomarkers and therapeutic targets. Targeting these pathways may help overcome chemoresistance and improve treatment outcomes in cancer patients.

## 1. Introduction

Doxorubicin (DOX) is an anthracycline antibiotic that plays an important role in cancer therapy due to its broad-spectrum activity against various malignancies, including breast, ovarian, and lung cancer, Hodgkin’s and non-Hodgkin’s lymphoma, leukemia, neuroblastoma, and sarcomas [1,2,3,4,5,6,7,8]. The primary mechanism of its action is intercalation into DNA with subsequent inhibition of topoisomerase II and double strand DNA repair, preventing the replication of rapidly dividing cancer cells [9]. Furthermore, DOX stimulates the generation of reactive oxygen species, which inflicts oxidative damage to cellular components, thereby augmenting its cytotoxic effects [10]. These combined action leads to cell cycle arrest and apoptosis, making DOX highly effective in combating tumor cells [11]. Despite its efficacy, the emergence of tumor cell resistance to DOX represents a significant complication in the treatment of diverse malignancies with this agent, potentially limiting its long-term success [8,12]. Furthermore, initial resistance to DOX can result in cross-resistance to structurally distinct drugs (also known as multidrug resistance (MDR) phenomenon), which markedly diminishes the efficacy of chemotherapy [13,14].

While several molecular mechanisms of DOX resistance have been identified, including the overexpression of ABC transporters, topoisomerase II alterations, and enhanced DNA repair [12,15,16,17,18], the comprehensive transcriptomic landscape governing this resistance remains poorly defined. The majority of previous studies have concentrated on downstream effectors rather than the underlying regulatory program or universal marker genes of resistance [19,20,21]. Consequently, a critical gap exists in our understanding of how DOX resistance emerges and is maintained across different tumor types. This knowledge gap significantly hampers both the prediction of therapeutic response and the development of effective strategies to overcome MDR.

Transcriptomics and computational biology offer powerful tools for unraveling the complex regulatory programs involved in different biological processes, enabling the identification of core regulators and shared molecular signatures across different models [22,23,24]. Although these approaches have been successfully applied to identify key regulators of resistance to various chemotherapeutics, including platinum compounds and taxanes [25,26,27,28,29,30,31], DOX resistance remains underexplored using such integrative methods. To the best of our knowledge, only a few studies have identified master regulatory genes associated with DOX resistance using bioinformatics approach [32,33,34,35]. However, these studies have been conducted on a single cancer type, and, in some cases, without proper experimental validation. Consequently, the identification of core transcriptomic signatures and master regulatory genes that confer DOX resistance across multiple tumor types is an urgent and unresolved challenge.

In this study, we performed a comprehensive re-analysis of independent, publicly available transcriptomic datasets from DOX-resistant cancer cells of different origins, which identified a set of genes associated with DOX desensitization and revealed their involvement in key biological processes. Following the reconstruction and topological analysis of gene association networks, we identified cross-cancer hub genes that form a core regulatory module strongly linked to DOX resistance. The significance of these hub genes was subsequently verified using a relevant in vitro model and corroborated by multiple bioinformatics approaches. Thus, our findings demonstrate the existence of a cross-cancer regulatory cluster underlying chemoresistance in cancer cells. The identified hub genes may represent promising novel biomarkers with potential clinical utility for predicting DOX sensitivity.

## 2. Materials and Methods

### 2.1. In Silico Analysis

#### 2.1.1. Data Collection and Identification of DEGs

Previously published cDNA microarray profiles associated with DOX resistance in cancer cells were obtained from the Gene Expression Omnibus (GEO) database (https://www.ncbi.nlm.nih.gov/geo/, accessed on 1 February 2024) using the keywords «doxorubicin resistance» and «cancer cells» (Table 1). Datasets were included in analysis if they met the following criteria: (i) included both doxorubicin-resistant and corresponding doxorubicin-sensitive (parental) control cells, (ii) contained at least two biological replicates per group, (iii) were derived from human cancer cell lines, and (iv) represented different cells origin. Datasets not meeting these criteria were excluded. For each dataset, the differentially expressed genes (DEGs) between the experimental (resistant) and control cell lines were identified using the GEO2R web service [36]. Adjusted *p*-values were calculated using the Benjamini–Hochberg false discovery rate method. An FDR-adjusted *p*-value < 0.05 and |fold change| > 1.5 were defined as the threshold for DEGs selection for further analysis. Subsequently, the overlapping of DEGs obtained from different datasets was conducted using the InteractiVenn tool (https://www.interactivenn.net/, accessed on 3 February 2025) [37]. The data were visualized as heatmaps using the Morpheus tool (https://software.broadinstitute.org/morpheus, accessed on 5 February 2025).

#### 2.1.2. Functional Analysis of DEGs

Functional annotation of DEGs was performed using the ToppFun server (Gene Ontology (GO) terms) (https://toppgene.cchmc.org/enrichment.jsp, accessed on 7 February 2025) [42]. To identify biological processes associated with DOX resistance, functional enrichment was carried out separately for each dataset. The lists of significantly enriched GO terms (adjusted *p*-value < 0.05) were then compared across datasets, and only terms consistently enriched in all three datasets were retained as cell context–independent processes. Dataset-specific terms that did not overlap were considered context-dependent and were excluded from further comparative analysis. To reduce redundancy the overlapping GO terms were clustered by semantic similarity using the REVIGO tool (http://revigo.irb.hr/, accessed on 8 February 2025) [43]. The functional annotation of the core genes associated with DOX resistance and their common first neighbors from each dataset was performed using the ClueGO 2.5.7 plugin [44] in Cytoscape 3.10.3 using the GO, Kyoto Encyclopedia of Genes and Genomes, WikiPathways, and REACTOME databases. The parameters were set as following: the GO Tree interval 3–8, the minimum number of genes per cluster − 3, *p*-value of enriched terms < 0.05, the GO Term Fusion was switched on, kappa score ≥ 0.4.

#### 2.1.3. Gene Association Networks Reconstruction and Topological Analysis

Gene association networks for each dataset were reconstructed using the STRING plugin (v2.2.0) (Search Tool for the Retrieval of Interacting Genes/Genomes) database [45] with a confidence score of greater than 0.7. The resulting networks were visualized using the Cytoscape 3.10.3 software. The interconnectivity score (degree) of common DEGs was calculated using the NetworkAnalyzer plugin (v4.5.0) [46] for each dataset and visualized using the Morpheus tool (https://software.broadinstitute.org/morpheus, accessed on 17 March 2025). The Molecular Complex Detection (MCODE) plugin (v2.0.3) [47] in Cytoscape 3.10.3 was used to identify the most interconnected regions (clusters) within the gene networks. Only the top five clusters, according to the MCODE score, were selected for further analysis.

#### 2.1.4. Drug Sensitivity Analysis

To evaluate the Pearson correlation between the expression level of marker genes and drug sensitivity, we utilized the Gene Set Cancer Analysis database (http://bioinfo.life.hust.edu.cn/GSCA/, accessed on 15 April 2025) [48] with data from the Cancer Therapeutics Response Portal. A positive correlation coefficient (r > 0.2) indicates drug resistance in cell lines with high gene expression, while negative coefficient (r < −0.2) indicates drug resistance in cell lines with low gene expression. Only correlations with FDR-adjusted *p*-values of less than 0.05 were considered as statistically significant.

#### 2.1.5. Text Mining Analysis

The co-occurrence of hub genes symbols with multidrug-associated terms in the abstracts of published articles from the MEDLINE database was performed using the GenCLiP3 tool (http://cismu.net/genclip3/analysis.php, accessed on 18 April 2025) [49] with the default settings followed by the visualization of obtained data using the Circos tool (https://circos.ca/, accessed on 20 April 2025) [50].

#### 2.1.6. Survival Analysis

The hazard ratio (HR) values of genes and gene signatures expression for patients diagnosed with various tumor types in The Cancer Genome Atlas (TCGA) database were assessed using the GEPIA2 web tool (http://gepia2.cancer-pku.cn/, accessed on 15 May 2025) [51]. Only HR values with *p*-values of less than 0.05 were considered statistically significant. Kaplan–Meier plots of overall patient survival were also obtained using GEPIA2 for the most hazardous gene signatures.

Kaplan–Meier plots for BRCA patients treated with DOX were constructed according to TCGA data using survminer (v0.5.1) package in R-4.5.1.

### 2.2. In Vitro Experiments

#### 2.2.1. Cell Cultures

Human cervical carcinoma KB-3-1 cells were purchased from the Russian Culture Collection (Institute of Cytology of the Russian Academy of Sciences, Saint Petersburg, Russia), and its subtype with MDR phenotype KB-8-5 cells were generously donated by Prof. M. Gottesman (National Institutes of Health, Bethesda, MD, USA). The cells were grown in Dulbecco’s Modified Eagle’s Medium (DMEM) (Sigma-Aldrich, St. Louis, MO, USA) containing 10% (*v*/*v*) heat-inactivated fetal bovine serum (Dia-M, Moscow, Russia), 10,000 IU/mL penicillin, 10,000 μg/mL streptomycin, and 25 μg/mL amphotericin (MP Biomedicals, Illkirch-Graffenstaden, France). KB-8-5 cells were cultured in the presence of vinblastine at 300 nM. The cells were incubated at 37 °C in 5% CO_2_.

#### 2.2.2. Cell Viability Assay

KB-3-1 and KB-8-5 cells were seeded at density of 1 × 10^4^ cells/well in 96-well plates in DMEM with 10% fetal bovine serum for 24 h. Then, the medium was replaced with serum-free medium containing DOX (1–10 μM) (D1515; Sigma-Aldrich, St. Louis, MO, USA) or without it (control). After the 72 h of incubation, 10 μL of MTT solution (10 μL, 5 mg/mL) (Sigma-Aldrich, St. Louis, MO, USA) were added to each well for 2 h. Finally, the formazan crystals were dissolved in DMSO, and the optical density was measured at 570 and 620 nm using a Multiscan RC plate reader (Thermo LabSystems, Helsinki, Finland).

#### 2.2.3. Doxorubicin Accumulation Assay

The intracellular accumulation of DOX in KB-3-1 and KB-8-5 cells was measured based on the intrinsic fluorescence of DOX (excitation and emission wavelengths of ~480 and ~590 nm, respectively), as previously described [52,53]. Briefly, the cells were incubated with 2 µM DOX for 2 h, washed twice with PBS to remove extracellular drug, and analyzed using a NovoCyte flow cytometer (ACEA Biosciences, San Diego, CA, USA). DOX fluorescence was detected directly in the cells, providing a quantitative measurement of intracellular drug accumulation. A total of 10,000 events were collected for each sample.

#### 2.2.4. Quantitative Real-Time PCR (qRT-PCR)

Total RNA was isolated from KB-3-1 after incubation with or without 1.5 µM of DOX for 6 or 48 h and KB-8-5 cells using the TRIzol reagent (Ambion, Austin, TX, USA) according to the manufacturer’s protocol. The first cDNA strand was synthesized from total RNA in the reaction mixture (100 μL) containing 20 μL of 5× RT buffer (Biolabmix, Novosibirsk, Russia), 250 U of M-MuLV-RH-revertase (Biolabmix, Novosibirsk, Russia), 4 μg of total RNA, and 100 μM of the random hexa-primers. Reverse transcription was conducted at 42 °C for 60 min followed by a denaturation step at 70 °C for 10 min. Afterward, cDNA amplification was carried out using the following protocol: an initial denaturation step at 94 °C for 5 min with 40 subsequent 40 cycles of amplification (95 °C for 30 s, 59 °C for 30 s, and 72 °C for 30 s). The final reaction mixture for amplification (25 μL/probe) included 12.5 μL of HS-qPCR (2×) master mix (Biolabmix, Novosibirsk, Russia), 5 μL of cDNA, and 0.25 μM of forward and reverse primers (Table 2). The relative expression of genes was normalized to *GAPDH* and calculated by the ΔΔCt method using the CFX96 Real-Time system (Bio-Rad, Hercules, CA, USA).

#### 2.2.5. Cholesterol Staining

KB-3-1 and KB-8-5 cells at density of 5 × 10^5^ cells/well were allowed to attach to glass coverslips in 24-well plates in serum-free DMEM for 24 h. After incubation, the cholesterol level in the cells was measured using a cholesterol assay kit (ab133116, Abcam, Cambridge, MA, USA) according to manufacturer’s instructions. Images were captured using a LSM710 confocal microscope (Zeiss, Oberkochen, Germany) at 40× magnification. The cholesterol level was calculated as mean cell fluorescence intensity (300 cells per group were analyzed in total) using ImageJ (v2.14.0) software.

#### 2.2.6. Cell Adhesion Assay

Cell adhesion was assessed using a trypsin assay, as previously described [54]. KB-3-1 and KB-8-5 cells were seeded at a density of 2 × 10^4^ cells/well in 96-well plates in DMEM with 10% fetal bovine serum (Dia-M, Moscow, Russia) and were allowed to adhere overnight. The cells were then washed with PBS and treated with 1:16 to 1:2 dilutions of TrypLE Express (Gibco, Grand Island, NY, USA) in PBS for 5 min at 37 °C. The plates were then washed three times with PBS to remove unattached cells, and the adherent cells were quantified using the MTT assay.

#### 2.2.7. Cell Morphology Assay

KB-3-1 and KB-8-5 cells were seeded into 12-well plates in triplicates at 2 × 10^5^ cells/well and incubated overnight. An EVOS XL Core microscope with CMOS camera (Thermo Fisher Scientific, Waltham, MA, USA) was then used to assess cell morphology at ×20 magnification. Aspect ratio (AR, the ratio of the major to minor axes) and cell area were quantified for random 250 cells for each cell line using ImageJ (v2.14.0) software.

#### 2.2.8. Cell Proliferation Assay

To assess cell proliferation rate, KB-3-1 and KB-8-5 cells were seeded into 12-well plates in triplicates at 1 × 10^5^ cells/well and incubated for 24 h. Thereafter, cell density was measured using Automated Cell Counter TC20 (Bio-Rad, Singapore) for each well.

### 2.3. Statistical Analysis

Statistical analysis of obtained data was conducted using the following methods: the Benjamini–Hochberg false discovery rate method (GEO2R tool, drug sensitivity analysis), a hypergeometric test with Bonferroni correction (ClueGO plugin, ToppFun tool), and a two-tailed unpaired Student’s *t*-test (qRT-PCR analysis, adhesion assay, cholesterol level assessment). The Shapiro–Wilk test was used to evaluate the normality of the data distribution in the in vitro studies. A *p*-value of less than 0.05 was considered statistically significant.

## 3. Results

### 3.1. Transcriptome Response Accompanying Doxorubicin Desensitization of Cancer Cells

To characterize the transcriptomic changes associated with the acquisition of doxorubicin (DOX) resistance by cancer cells, three previously published independent cDNA microarray datasets obtained from DOX-resistant cancer cells of different origins, including leukemia, neuroblastoma, and breast cancer cells [38,39,40,41], were analyzed. Comparing the transcriptomes of DOX-resistant cells and their drug-susceptible counterparts revealed a similar numbers of differentially expressed genes (DEGs) in neuroblastoma cells (1076 up-regulated and 1071 down-regulated genes) and leukemia cells (1181 up-regulated and 1721 down-regulated genes), whereas breast cancer cells were more susceptible to DOX desensitization (1943 up-regulated and 2061 down-regulated genes) (Figure 1A). Interestingly, despite these differences, a range of identical genes were revealed among the most changed DEGs in these cells, including *ABCB1* and *ABCB4* in leukemia and breast cancer cells (encode known MDR-associated proteins), as well as *VIM* in breast cancer and neuroblastoma cells (encodes a cytoskeletal protein crucial for epithelial–mesenchymal transition (EMT), a process tightly associated with cancer drug resistance [55]) (Figure 1A). These data, along with the upregulation of other EMT-related DEGs in breast cancer cells (*CDH1* and *MMP1*, which encode N-cadherin and matrix metalloproteinase 1, respectively) and the downregulation of *ALOX5* (which plays a drug-sensitizing function [56]) in leukemia cells, indicate the credibility of obtained results.

A more detailed examination of the top 10 most activated and suppressed DEGs in analyzed cells revealed groups of genes involved in similar processes, such as cell adhesion (*AGR2*, *CDH1*, *CTGF*, *GJA1*, *ITGAM*, *KRT19*, *VIM*), differentiation (*CRABP1*, *DLK1*, *FSTL1*, *LRCC17*, *MECOM*), proliferation (*CDKN1A*, *CILP*, *LINC01419*, *NT5DC4*, *TSPAN9*), energetics (*GLUL*, *LDHB*), immune response (*B2M*, *COLEC11*, *FCER1A*, *FGL2*, *SRGN*), and signal transduction (*CNRIP1*, *GRK5*, *HEY1*, *IGF1*, *S100A10*, *S100P*, *SOCS2*, *TCEAL7*, *UBB*, *UCHL1*) (Figure 1A). These data clearly indicate multiparametric changes in cellular phenotype during the acquisition of chemoresistance and a similarity in the response of different cancer cells to DOX desensitization. Indeed, despite their diversity, the Gene Ontology (GO) terms most representative for the analyzed DOX-resistant cells are related to tightly interconnected processes such as cell morphology, cell adhesion and underlying cytoskeleton reorganization (Figure 1B). Additionally, the acquisition of breast cancer and neuroblastoma cells with DOX chemoresistance was also accompanied by changes in the expression of genes involved in cell viability and proliferation (Figure 1B).

To identify cell context–independent processes, we overlapped the GO terms significantly enriched in each dataset. Only those processes consistently enriched across all three cancer cell types were retained, while non-overlapping, dataset-specific terms were considered context-dependent and excluded from downstream analysis. This approach yielded 147 processes that were shared among the three models (Figure 2A). Their further grouping according to function similarity demonstrated high enrichment of terms related to cell apoptosis, cell cycle, transcription, and cell adhesion-dependent processes such as cell migration and tube development (Figure 2B).

Taken together, these data clearly demonstrate that the acquisition of DOX resistance by cancer cells is accompanied by similar transcriptome changes, regardless of cell origin. These changes help the cells to overcome the toxic effects of DOX by increasing cell viability and reducing the xenobiotic load through changes in cell adhesiveness and motility.

### 3.2. Reconstruction of DOX Resistant-Associated Gene Networks and Their Module Analysis

To evaluate possible functional relationships among the revealed DEGs, gene association networks were reconstructed for each analyzed dataset using the STRING database. As shown in Figure 3A, all gene networks had a similar architecture, with a densely packed center and weakly connected genes on the periphery. Consistent with the more pronounced transcriptomic response of breast cancer cells to DOX desensitization (Figure 1A), the breast cancer network was denser than those of neuroblastoma and leukemia, as reflected by their node/edge ratios (4.7, 2.2, and 1.8, respectively). Since functionally related genes can form tightly interconnected clusters [57], we performed a cluster analysis of the reconstructed gene networks. The top five gene clusters with the highest normalized enrichment scores for each dataset are depicted in Figure 3B. Despite the different seed genes, the revealed gene clusters included genes common to different cell types. For example, cluster pairs 2-2 and 3-5 contained four and five common genes, respectively, between breast cancer and neuroblastoma cells (Figure 3C). These results support our hypothesis of a conserved transcriptomic core associated with DOX resistance.

Interestingly, the functional annotation of the isolated gene clusters did not reveal terms directly linked to the aforementioned cell viability and adhesion-related processes. However, it showed high enrichment of functional groups linked to more subtle processes underlying the cellular stress response, such as cell energetics, cholesterol biosynthesis, RNA processing, ribosome biogenesis, regulation of translation, and proteasomal degradation (Figure 3D). As expected, gene clusters related to different cells were characterized by obvious similarity in their functional relationships (Figure 3E). Specifically, several molecular processes, including cellular respiration and RNA processing, were identified as common to gene clusters from different cells (Figure 3E).

### 3.3. Identification of Core Genes Associated with DOX Resistance

The results of the functional analysis clearly indicated that the resistance of tumor cells to DOX is associated with specific changes in the functional landscape, which may suggest the existence of a regulatory core responsible for this response. To identify genes tightly associated with DOX desensitization of cancer cells, the activated and suppressed DEGs from three analyzed datasets were distinctly overlapped, revealing a core gene group comprising 16 upregulated and 21 downregulated genes (Figure 4A). Hierarchical clustering of these genes according to their expression patterns revealed two main clades of activated and suppressed genes and one out-group consisted of the most overexpressed DEGs (Figure 4B). This out-group consisted of *ABCB1*, which encodes the efflux transporter P-glycoprotein, a well-known inducer of multidrug resistance, and *GJA1*, a gap junction component involved in chemoresistance regulation of glioblastoma cells [58]. Other core genes with the most changed expression levels were overexpressed *ZYX* and *TUBA4A*, which are involved in cell adhesion and cytoskeleton organization, respectively, and downregulated *MAP7* and *CD24*, encoding microtubule-stabilizing protein ensconsin and immune-related signal transducer CD24, respectively.

### 3.4. Identification of Key Biological Processes Associated with Chemoresistance Core Genes

To understand the processes in which the revealed core genes participate, we retrieved their first neighbors from the reconstructed regulomes (Figure 2A) and performed a functional analysis of the resulting core gene-centered subnetworks. As shown in Figure 4C, the activated core genes were associated with gap junction, mTORC1 and Wnt signaling pathways, regulation of transcription, and ribosome biogenesis. In turn, the suppressed core genes were involved in regulation of cell cycle progression, steroid biosynthesis and Gβγ signaling pathway (Figure 4C). More detailed analysis of the obtained functional map revealed that chemoresistance of cancer cells was associated with enhanced transport of connexons to plasma membrane resulting in more effective gap junction formation (Figure 4D), which may led not only to enhanced cell-to-cell communications, but also cell–cell adhesion [59]. Additionally, desensitization of cancer cells to DOX led to G1 phase perturbations and weakened cholesterol biosynthesis (Figure 4D).

To verify these findings, we used the chemoresistant human cervical carcinoma KB-8-5 cells and their drug-sensitive parental KB-3-1 cells. We confirmed that KB-8-5 cells exhibit significant DOX resistance (Figure 4E), which was primarily attributed to a 37.4-fold reduction in intracellular drug accumulation compared to KB-3-1 cells (Figure 4F, lower panel). Consistent with our previous results, KB-8-5 cells demonstrated stronger adhesive properties compared to KB-3-1 cells, showing a twofold difference under the most aggressive trypsinization conditions (Figure 4G). Furthermore, Filipin III staining revealed significantly lower cholesterol content in KB-8-5 cells than in KB-3-1 cells (Figure 4H), suggesting suppressed cholesterol biosynthesis in chemoresistant cells.

In addition to these characteristics, KB-8-5 cells exhibited other distinct phenotypic features compared to KB-3-1 cells. Specifically, they were larger (Figure 5A–D), displayed a more elongated, spindle-like morphology, as shown by an increased aspect ratio (major-to-minor axis length) (Figure 5A,E) and demonstrated a lower proliferation rate (Figure 5F). These morphological data are consistent with the identified activation of EMT-related genes in DOX-resistant cells (Figure 1A).

Taken together, these findings demonstrate the reliability of our bioinformatics-based data and confirm that the acquisition of chemoresistance by cancer cells is accompanied by obvious phenotypic alterations.

### 3.5. Identification of Hub Genes Among DOX Resistant-Related Core Genes

At the next stage of the study, topology analysis of DOX resistance-related regulomes of cancer cells (Figure 2A) was performed to assess the nodal position of the identified core genes, as their potential master regulatory role in desensitization of cancer cells to chemotherapeutics may be indicated by this information [60]. As shown in Figure 6A, the genes most strongly associated with DOX resistance were regulators of the cell cycle (*CCND1* and *SEH1L*), ribosome biogenesis (*HEATR1*), intracellular signaling (*PLCB1*), and cell-to-cell communication and adhesion (*GJA1*). To prioritize diagnostically relevant candidates, the hub characteristics of these genes were further balanced with their expression levels, and the most activated ones among the hub genes were selected (Figure 6B). In addition to the aforementioned *SEH1L* and *GJA1*, this list included genes encoding a transcription factor associated with mesenchymal–epithelial transition (*TCF3*), tubulin alpha (*TUBA4A*), and zyxin, which plays a crucial role in focal adhesion (*ZYX*).

Next, to determine the extent to which these genes have been studied in the context of cancer chemoresistance, a text mining analysis was performed using the following keywords: “drug resistance,” “doxorubicin,” “chemotherapy,” “reduced sensitivity,” and “cancer therapy.” An analysis of the MEDLINE database revealed that only *GJA1* demonstrated high co-localization with the aforementioned keywords within article abstracts, while *TCF3*, *ZYX*, *TUBA4A*, and *SEH1L* were mentioned in the context of chemoresistance to a much lesser extent (Figure 6C), which may suggest that they can be considered as novel marker candidates for DOX resistance.

### 3.6. Verification of Hub Genes In Silico and In Vitro

To further verify the activation of the revealed hub genes in tumor cells with drug-resistant phenotypes, the correlation between their overexpression with low sensitivity of tumor cells to commonly used chemotherapy drugs was estimated using the Cancer Therapeutics Response Portal. It was found that high expression of *ZYX*, *GJA1*, and *TUBA4A* indeed correlated well with low sensitivity of tumor cells to DOX and, moreover, to another anticancer drugs, such as vincristine, topotecan, panobinostat, and cytarabine (Figure 6D). Surprisingly, the overexpression of *SEH1L* and *TCF3* was associated with vice versa high sensitivity of tumor cells to these compounds (Figure 6D). Considering these results, we speculated that the activation of *SEH1L* and *TCF3* observed in DOX-resistant tumor cells (Figure 4B) may be a consequence of compensatory mechanisms in response to xenobiotic stress. We hypothesize that cells more sensitive to chemical treatment may be characterized by greater activity of the NUP107-160 nuclear pore subcomplex and EMT-related cellular plasticity, in which these genes play important roles [61,62]. Despite this discrepancy, *SEH1L*, similar to *ZYX*, *GJA1*, and *TUBA4A*, was upregulated in DOX-resistant breast cancer cells in the independent transcriptomic dataset GSE76540 (Figure 6E). Subsequent RT-PCR analysis confirmed the upregulation of *ZYX*, *TUBA4A*, *SEH1L*, and *TCF3* in DOX-resistant KB-8-5 cells by 3.4-, 2.3-, 2.3-, and 2.0-fold, respectively, compared with DOX-sensitive KB-3-1 cells (Figure 5F). *GJA1* also tended to be activated in KB-8-5 cells; however, this change was not statistically significant (Figure 6F). Additionally, to determine whether these gene expression changes reflect early adaptive responses or stable resistance mechanisms, we further examined drug-sensitive KB-3-1 cells after 6 h and 48 h exposure to a semi-toxic concentration of DOX (1.5 µM) (Figure 4E). Neither short- nor long-term treatment altered the expression level of core genes (Figure 6G), indicating that their upregulation represents established, late-stage resistance mechanisms rather than transient drug-induced effects.

Taken together, our results confirm that the activation of the identified hub genes is consistently associated with DOX desensitization in cancer cells. The limited prior association of *SEH1L*, *TUBA4A*, *ZYX*, and *TCF3* with chemoresistance (Figure 6C), combined with their consistent dysregulation across cancer types (Figure 4B and Figure 5D,F), nominates them as novel and promising biomarker candidates for predicting DOX sensitivity.

### 3.7. Impact of DOX Resistance-Related Marker Genes on the Survival of Cancer Patients

Given that the activation of genes related to drug resistance leads to reduced sensitivity to chemotherapy and poor prognosis in cancer patients [63], we questioned whether the expression of hub genes associated with overall patient survival. To address this issue, we calculated hazard ratio (HR) values for *SEH1L*, *TUBA4A*, *TCF3*, *ZYX*, and *GJA1*, both individually and in combination, using tumor data from The Cancer Genome Atlas (TCGA). Statistically significant hazard ratio (HR) values were identified for several cancer types, including cervical squamous cell carcinoma (CESC), hepatocellular carcinoma (LIHC), sarcoma (SARC), prostate adenocarcinoma (PRAD), skin cutaneous melanoma (SKCM), breast carcinoma (BRCA), and thymoma (THYM), as shown in Figure 7A.

We found that the overexpression of hub genes correlates well with poor cancer patient outcomes. PRAD demonstrated the most pronounced HR values among other malignancies. The most hazardous correlations were shown with *TCF3* (HR = 7.1) and the gene signatures *SEH1L* + *TUBA4A* + *ZYX* (HR = 4.8) and *SEH1L* + *TUBA4A* + *ZYX* + *TCF3* (HR = 5.1) (Figure 7B). Consistent with the aforementioned results, BRCA and CESC, which were used to identify and experimentally validate core genes, respectively (Figure 4A and Figure 5F), also demonstrated high HR values for *SEH1L* + *TUBA4A* (HR = 1.7) (Figure 7C) and *GJA1* + *ZYX* (HR = 2.5) (Figure 7D). As shown in Figure 7A, LIHC and CESC revealed the highest number of correlations with analyzed gene signatures (19 and 12 out of 31, respectively). Interestingly, the overexpression of hub genes in THYM and SKCM was associated with higher overall survival rather than poor outcomes, which can be explained by the peculiarities of treating these tumor types.

Finally, to evaluate the association between the core genes and overall survival in cancer patients treated with DOX, we focused our analysis on the BRCA cohort. From this cohort, we selected 363 patients who received DOX, doxorubicin hydrochloride, or their liposomal forms. Other patient cohorts were excluded from this analysis due to fragmentary information in the TCGA database regarding the inclusion of DOX in their treatment regimens. As shown in Figure 7E, overexpression of *SEH1L*, *TCF3*, *SEH1L* + *TCF3* combination, and *SEH1L* + *TCF3* + *TUBA4A* and *SEH1L* + *TCF3* + *ZYX* gene sets was associated with a poorer prognosis in DOX-treated BRCA patients. These findings confirm that the activation of several core genes, primarily *SEH1L* and *TCF3*, is associated with lower survival rates following DOX treatment. However, further analysis of their expression in larger cohorts of DOX-treated patients is necessary to definitively confirm their diagnostic potential.

Interestingly, we found that *SEH1L* and *TCF3* activation was associated with increased chemosensitivity to DOX in vitro in the Cancer Therapeutics Response Portal database (Figure 6D), contrasting with our clinical findings. This discrepancy underscores the necessity of careful interpretation of in vitro drug response data when translating findings to a clinical context.

## 4. Discussion

Despite the broad clinical utility of DOX in treating a wide range of malignancies, its therapeutic efficacy is often undermined by the emergence of chemoresistant phenotypes accompanied by significant transcriptomic alterations. Understanding the mechanisms underlying this process is critical; however, only a few studies have conducted bioinformatics analyses of the transcriptome specifically associated with DOX desensitization exclusively in breast cancer cells [32,33,34,35,64]. These reports do not provide a comprehensive understanding of whether a cell-context-independent core gene set controls DOX resistance or if distinct regulatory networks drive this phenomenon in different cell lines.

To address this knowledge gap, we pursued two primary objectives in this study. First, we employed a comprehensive approach to functionally annotate DEGs and their clusters, enabling us to characterize a cell-context-independent functional landscape associated with DOX resistance. Second, we focused on identifying common hub genes within chemoresistant regulomes across cell lines of different origins and evaluating their potential as biomarkers for DOX resistance.

### 4.1. Identification of Key Processes Associated with DOX Resistance in Cancer Cells

Our findings revealed notable similarities in several biological processes in the analyzed cells following the loss of sensitivity to DOX, suggesting the existence of a conserved transcriptional program responsible for chemoresistance. Functional analysis of DEGs (Figure 1B, Figure 2 and Figure 4C) and their tightly interconnected clusters (Figure 3D,E) revealed that desensitization of tumor cells to DOX is associated with modulation of cell viability, reorganization of cytoskeleton (cell motility, cell adhesion, gap junction formation, and vasculogenic mimicry), and biosynthesis of cholesterol (Figure 1B, Figure 2 and Figure 4C), which are presumably associated with changes in cellular respiration, RNA processing, and the balance of protein expression and degradation (Figure 3D,E). These results characterize the activation of a complex adaptive program in response to chemotherapy-induced cell stress and are consistent with previously published data [65,66,67,68]. The link between therapy resistance in cancer and deregulation of apoptosis, cell cycle, and impaired actin dynamics has been thoroughly reviewed [67,69,70]. The revealed functional terms associated with cell energetics (Figure 3D) corroborate the findings of Ippolito et al. and Wu et al., who reported an association between increased oxidative respiration and resistance to DOX and docetaxel in hepatocellular and prostate carcinoma, respectively [65,71]. Changes in RNA processing and protein biosynthesis also play an important role in desensitization of tumor cells to DOX since abnormal alternative splicing is involved in developing drug resistance [72], and anabolic pathways—including protein biosynthesis—strongly influence cell proliferation and may contribute to the adaptive survival of tumor cells [66].

Since “gap junction” and “cholesterol biosynthesis” are significantly enriched functional terms in DOX-resistant cells (Figure 4C,D) and are less characterized in the context of MDR, we conducted a series of experiments to confirm the phenotypic changes associated with these processes in DOX-resistant KB-8-5 cells.

In the case of gap junctions, we did not focus on their constitutive function (intercellular communication), but rather on their connection with cell adhesion, which is described in detail in a recent review by Lucaciu et al. [73]. We were also prompted to make this decision by the described effect of connexins, the pivotal components of gap junctions, on cell adhesion-associated β-tubulin dynamics and cell stiffness, which, as reported by Fu et al. [74] and Han et al. [75], mediates resistance of breast and colorectal carcinoma cells to paclitaxel and 5-fluorouracil, respectively. In line with this, we found that DOX-resistant KB-8-5 cells exhibited significantly greater adhesiveness than chemosensitive KB-3-1 parenteral cells (Figure 4G), directly linking enhanced cell adhesion to DOX resistance. These data are consistent with the identification of a group of adhesion-related hub genes in DOX-resistant cells, including *ZYX*, *GJA1*, and *TUBA4A*, which play a direct role in cell adhesion [59,76,77,78], as well as *TCF3* and *SEH1L*, which control cell adhesive properties through a variety of protein mediators [79,80] (Figure 6).In the case of cholesterol biosynthesis, filipin III staining demonstrated significantly reduced cholesterol content in DOX-resistant KB-8-5 cells compared to parental KB-3-1 cells (Figure 4H). This finding is consistent with our functional analysis, which showed predominant suppression of cholesterol biosynthesis-related genes (Figure 4C). These findings are consistent with recent work of Criscuolo et al., who reported that platinum resistance in ovarian cancer cells was associated with downregulation of farnesyl diphosphate synthase and oxidosqualene cyclase, key cholesterol biosynthesis enzymes [81]. Despite the observed downregulation, Criscuolo et al. demonstrated that chemoresistant tumor cells exhibited elevated cholesterol levels due to its active extracellular uptake [81]. Since filipin III staining was conducted in serum-free DMEM, the uptake of extracellular lipids by the model cells was ineffective. This could explain the reduction in cholesterol content observed in DOX-resistant KB-8-5 cells (Figure 4H). Considering that cholesterol forms an optimal lipid composition for correct P-glycoprotein function when incorporated into the cell membrane [82], it appears that cholesterol accumulation in lipid rafts is more important than overall cholesterol synthesis for developing drug resistance. Since filipin III staining only shows the distribution of free cholesterol within cells (Figure 4H), further studies on the content of membrane-associated cholesterol in DOX-resistant cells are necessary.

Collectively, the first part of our study confirmed the existence of a conserved functional landscape associated with DOX resistance across diverse tumor cell types. The observed similarity in cellular response to DOX desensitization suggests the presence of a shared regulatory network underlying this phenomenon. Based on this finding, our research was further focused on identifying the core genetic determinants of DOX resistance.

### 4.2. Identification of Core Genes Associated with DOX Resistance in Cancer Cells

A comparative analysis of DEGs across all analyzed DOX-resistant cells revealed a regulatory core consisting of 16 upregulated and 21 downregulated genes (Figure 4A,B). The presence of well-established chemoresistance markers in this core set, including *ABCB1* (which encodes P-glycoprotein, a known MDR transporter) [83,84,85], as well as *CD24* (a modulator of chemosensitivity to DOX, cisplatin, and paclitaxel) [86,87], confirms the reliability of our results obtained via a bioinformatics approach.

A more detailed analysis of core genes, including their ranking according to interconnections within reconstructed regulomes and expression levels, revealed five genes (*GJA1*, *TCF3*, *ZYX*, *TUBA4A*, and *SEH1L*) with high degree scores and pronounced activation (Figure 6B), which allows us to considered them as promising marker candidates of DOX resistance. According to TCGA data, these genes had high prognostic potential, both individually and as part of gene signatures, and demonstrated significant hazard ratio values associated with cancer patient mortality (Figure 7). Verification using independent transcriptomic data (Figure 6D,E) and RT-PCR analysis of KB-8-5 and KB-3-1 cells (Figure 6F) confirmed the upregulation of these genes in chemoresistant cells. Text mining revealed that only *GJA1*, which encodes connexin 43 (a component of gap junctions), has been extensively evaluated in the drug resistance field. Other hub genes have been investigated much less in this context (Figure 6C). To elucidate their potential roles in DOX desensitization, we summarize below the published evidence linking them to the regulation of cell survival process.

*GJA1* encodes connexin 43 (Cx43), which is a component of gap junctions involved in intercellular communication. Although gap junctions have been considered tumor suppressors due to their role in maintaining tissue homeostasis through cell–cell communication [88], recent studies have revealed their tissue-dependent tumorigenic functions, including drug resistance [89,90]. For instance, *GJA1* has been implicated in glioblastoma resistance to temozolomide by modulation of apoptosis [91,92] and activation of cytoprotective PI3K/Akt pathway [93]. Consistent with this, inhibition or knockdown of Cx43 has been shown to significantly restore the susceptibility of glioblastoma cells to temozolomide [91,94,95]. Several studies have demonstrated that GJA1 can translocate to mitochondria, protecting cardiomyocytes from death by stabilizing ion homeostasis, reducing reactive oxygen species, and preventing cytochrome c release [96,97]. Since the pharmacological effect of DOX is manifested through the induction of oxidative stress [10], we speculate that the observed *GJA1* overexpression in DOX-resistant tumor cells may be an attempt by cells to reduce reactive oxygen species production in response to xenobiotic stress.*ZYX* encodes zyxin, a protein essential for cell focal adhesions, which is associated with increased motility, adhesiveness and metastatic potential in several tumor types, including hepatocellular and colorectal carcinomas [76,98,99]. The role of *ZYX* in chemoresistance is unknown; however, Yang et al. recently revealed *ZYX* overexpression in taxol-resistant ovarian cancer cells [100], which is consistent with our findings. Apparently, the association of *ZYX* with DOX desensitization is related not only to its ability to control the restructuring of cellular cytoskeleton, but also its involvement in the Akt/mTOR signaling pathway, promoting cell survival and proliferation [98,101].TCF3 is E protein (class I) family of helix-loop-helix transcription factors that regulates cell differentiation [102,103]. TCF3 primarily activates the Wnt/β-catenin signaling pathway, which is a key driver of cancer cell stemness properties and EMT promotion [104,105], the processes contributing to tumor progression and therapeutic resistance [106]. *TCF3* silencing has been shown to impair the self-renewal capacity of breast cancer cells and reduce the expression of stemness-related genes [62]. Furthermore, Nie et al. showed that *TCF3* knockdown with siRNA decreased the expression of mesenchymal markers, subsequently suppressing the migration and invasion of uveal melanoma cells [102]. Given that EMT has been shown to markedly diminish the sensitivity of tumor cells to chemotherapeutic agents, including DOX [107,108], we propose that observed overexpression of *TCF3* in DOX-resistant cells (Figure 4B) may be linked to their partial mesenchymal phenotype. This is evidenced by the upregulation of *VIM* and downregulation *CDH1* in analyzed cells, both of which are known molecular markers of EMT (Figure 1A). Additionally, gene set enrichment analysis revealed significant overrepresentation of EMT-associated functional terms related to cell adhesion, migration and Wnt signaling in DOX-resistant tumor cells (Figure 1B, Figure 2B and Figure 4C).*TUBA4A* encodes tubulin alpha-4A, which has been associated with various neurodegenerative diseases [109,110,111], but has not been described yet as a MDR-associated gene. To the best of our knowledge, among tubulins, only the aberrant expression of tubulin beta-3 (*TUBB3*) has been shown to correlate with the resistance to paclitaxel in ovarian, breast and lung cancers [112,113,114] and to docetaxel in gastric cancer [115]. The observed overexpression of *TUBA4A* in DOX-resistant cells (Figure 4B) may be associated with enhanced cell proliferation in response to chemotherapy. This hypothesis is supported by Manissorn et al., who demonstrated that induced α-tubulin overexpression enhances proliferation, promotes mitosis, and confers protection against calcium oxalate-induced cell cycle disruption in myeloid-derived suppressor cells [116]. Because TUBA4A is naturally detyrosinated, it forms more stable microtubules than other tubulins [117]. This property may also play an important role in desensitization of tumor cells to chemotherapy.The protein product of the *SEH1L* gene is a component of the NUP107-160 nuclear pore subcomplex, which is essential for proper nuclear pore complex (NPC) function, normal kinetochore microtubule attachment, mitotic progression, and chromosome segregation [118,119,120]. NPC formation is closely associated with tumorogenesis [61] and NUP107-160 members, particularly NUP88 and NUP107, have been shown to promote the survival and invasion of cervical cancer cells [121]. Feng et al. demonstrated that *SEH1L* knockdown induces ferroptosis and suppresses hepatocellular carcinoma growth in vitro and in vivo [80]. Although NUP107-160, particularly *SEH1L*, is associated with tumorigenesis, its role in drug resistance remains unknown. Only NUP62 has been shown to affect drug sensitivity: its knockdown confers cisplatin resistance in ovarian carcinoma cells [122]. Given the key role of NPCs in the nuclear export of RNA and proteins, we hypothesize that *SEH1L* can potentially modulate the transport of MDR-related mRNA and regulatory RNAs from the nucleus, thereby mediating resistance to DOX.

As evidenced by the literature analysis presented, the identified core genes perform unique, non-overlapping functions. Nevertheless, the gene network reconstruction from the STRING database revealed that four out of the five core genes are interconnected (Figure 8). These associations are either direct, as in the case of *GJA1* and *TUBA4A*, or mediated by first-order gene neighbors, some of which possess a documented, albeit weak, link to DOX treatment or desensitization (Figure 8). As shown in Figure 8, *ZYX*, *GJA1*, and *TUBA4A*, which are all connected to cytoskeleton and adhesion regulation, form stronger interactions with each other compared to *TCF3*, a gene controlling the EMT-related transcriptional response. As expected, the nuclear pore-associated gene *SEH1L* is situated separately and does not form functional interactions with the other core genes (Figure 8). These findings suggest the presence of a multifaceted and, at least in part, coordinated transcriptomic response to prolonged DOX treatment that enables tumor cells to evade xenobiotic stress.

### 4.3. Limitations of the Study

Although our integrative bioinformatics analysis revealed promising candidates for cross-cancer DOX resistance markers, several limitations should be acknowledged. First, our data are primarily based on the transcriptomic profiling of DOX-resistant cell lines. Further study is required to validate the results using data from relevant animal models and clinical samples. Second, our validation experiments were limited to one chemoresistant cell line, KB-8-5 cells, and only the mRNA levels of the identified core genes were measured. Further expansion of these studies is needed, including studies on a large panel of DOX-resistant cells and analysis of the expression of the proteins encoded by core genes. Third, while we identified novel associations between certain genes and DOX resistance, mechanistic studies, including knockout experiments, are necessary to determine their precise role in chemoresistance. Lastly, to delineate the temporal activation of these genes during the acquisition of resistance, future work should involve generating resistant cell subline via long-term exposure to incrementally increasing DOX concnetrations and assessing core gene expression at serial time points.

## 5. Conclusions

Collectively, our data demonstrate that DOX desensitization in cancer cells of diverse origins is associated with a consistent set of transcriptomic alterations, indicating a shared regulatory core. Although a comprehensive characterization of this regulatory core remains incomplete, we have identified its pivotal component comprising the hub genes *GJA1*, *SEH1L*, *TCF3*, *TUBA4A*, and *ZYX,* which exhibit central regulatory roles and influence diverse pathways linked to chemoresistance, including experimentally validated perturbations in cell adhesion and cholesterol biosynthesis. Given their limited prior association with chemoresistance, further in-depth investigation of these hub genes is warranted. Such research could yield critical insights into the molecular pathogenesis of MDR and inform novel therapeutic strategies aimed for restoring chemosensitivity in cancer patients.

## Figures and Tables

**Figure 1 biomedicines-13-02527-f001:**
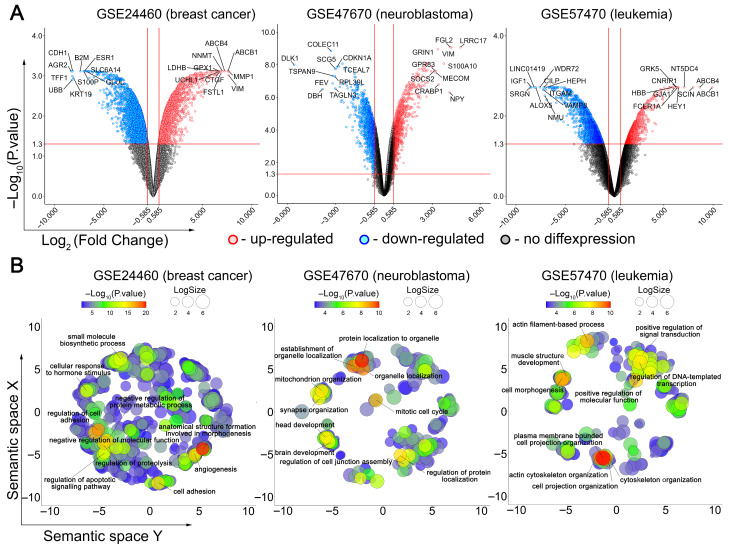
Transcriptomic response of cancer cells to DOX desensitization. (**A**) Volcano plots of differentially expressed genes (DEGs) (DOX-resistant versus DOX-susceptible cells) identified for three independent cDNA microarray datasets. The red, blue and grey dots represent the upregulated, downregulated and unchanged genes, respectively. The top 10 most up- and downregulated genes are signed. The horizontal and vertical red lines indicate the *p*-value and fold change thresholds, respectively (*p*-value < 0.05, |fold change| > 1.5). (**B**) Functional analysis of DEGs using Gene Ontology (GO) annotation. The text emphasizes the most significant terms identified through a semantic similarity analysis of enriched GO terms using REVIGO. LogSize indicates the number of functionally similar terms combined into one cluster.

**Figure 2 biomedicines-13-02527-f002:**
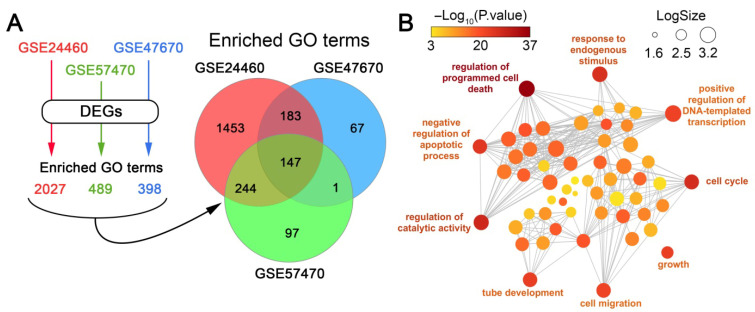
Identification of cell context-independent processes involved in the acquisition of DOX resistance by cancer cells. (**A**) Scheme illustrating the identification of enriched processes for each dataset and Venn diagram showing the overlap of significantly enriched GO terms (*p* < 0.05) among the analyzed datasets. (**B**) An interactive graph generated by REVIGO that illustrates the most enriched processes among the 147 GO terms that are common to all of the analyzed datasets.

**Figure 3 biomedicines-13-02527-f003:**
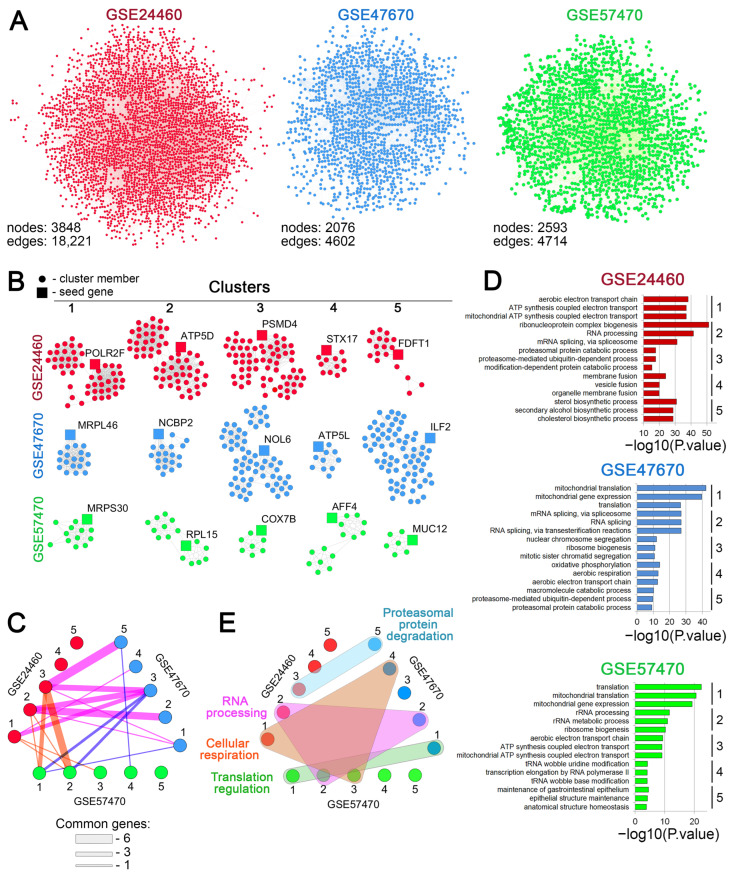
Cluster analysis of the regulomes of cancer cells associated with DOX resistance. (**A**) Gene networks reconstructed from differentially expressed genes (DEGs) using the STRING database (confidence score > 0.7). (**B**) Gene clusters identified within the reconstructed networks using the MCODE plugin in Cytoscape. The top five clusters with the highest enrichment score are depicted. (**C**) An interactive graph demonstrating the similarity of the identified MCODE clusters across the analyzed datasets. The edges represent the number of shared genes between clusters; line width is proportional to the number of common genes. (**D**) Functional annotation of the top five clusters’ genes, performed by the ToppFun tool. The numbers to the right of the bar plots indicate the cluster numbers. (**E**) A timing belt plot illustrating the common biological processes associated with the genes from the revealed MCODE clusters. Gene clusters from different datasets functionally enriched for the same biological processes are circled with colored shapes. Numbers in **C**–**E** correspond to the clusters indicated in **B** for each dataset.

**Figure 4 biomedicines-13-02527-f004:**
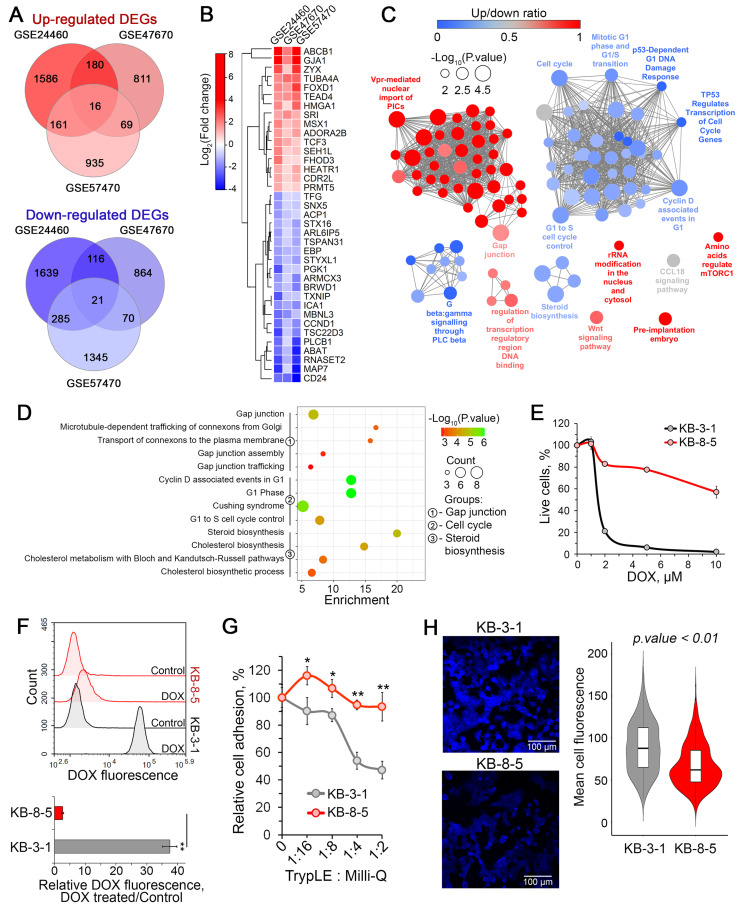
Identification of core genes associated with DOX resistance and their functional analysis. (**A**) Venn diagrams showing the overlap of differentially expressed genes (DEGs) that were either upregulated or downregulated in each analyzed dataset. (**B**) A heatmap showing the relative expression levels of core genes in DOX-resistant cell lines compared to DOX-sensitive parental cells. (**C**) Functional analysis of gene groups consisting of core genes related to DOX resistance and their first neighbors from the reconstructed gene networks depicted in Figure 3A. The analysis was performed using the Gene Ontology (biological processes), Kyoto Encyclopedia of Genes and Genomes, REACTOME, and WikiPathways databases via the ClueGO plugin in Cytoscape. (**D**) An enrichment bubble plot of functional terms linked with the most enriched processes, including gap junction, the cell cycle, and steroid biosynthesis. (**E**) Sensitivity of KB-8-5 and KB-3-1 cells to DOX (1–10 μM) evaluated by MTT after 72 h of incubation. (**F**) DOX fluorescence intensity in KB-8-5 and KB-3-1 cells after a two-hour incubation with 2 μM DOX, as measured by flow cytometry. Peak histogram of cell fluorescence (upper panel) and relative DOX fluorescence (lower panel). Control: cells incubated without DOX. (**G**) Adhesiveness of KB-8-5 and KB-3-1 cells. The cells were treated with various dilutions of TrypLE, then washed with PBS. The unattached cells were measured by an MTT assay. (**H**) Fluorescent staining of cholesterol in KB-8-5 and KB-3-1 cells using filipin III. The violin plot (right panel) shows the mean cell fluorescence of the analyzed groups, and representative images (left panel) are provided. ** *p* < 0.01, * *p* < 0.05.

**Figure 5 biomedicines-13-02527-f005:**
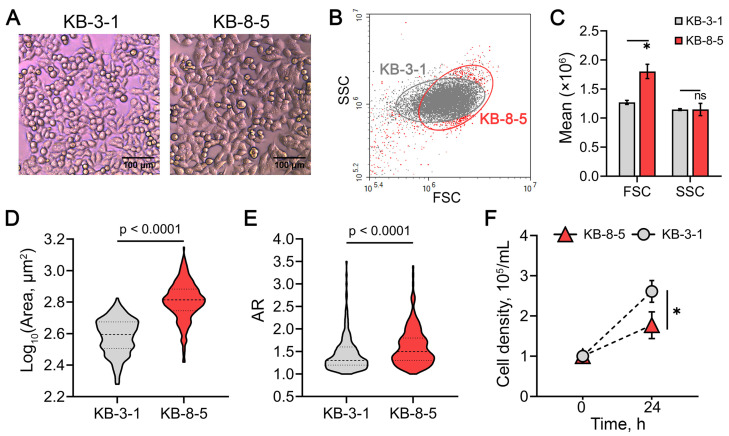
Phenotypic characterization of KB-3-1 and KB-8-5 cells. (**A**) Representative light microscopy images of the cells at ×20 magnification. (**B**) Cytogram of forward scatter (FSC) versus side scatter (SSC), indicating relative cell size and granularity, respectively. (**C**) Distribution histogram of FSC and SSC values. (**D**,**E**) Violin plots showing the distribution of cell size (**D**) and aspect ratio (AR) (**E**). Results of light microscopy analyzed by ImageJ tool. Dashed and dotted lines indicate the median and first and third quartile, respectively. (**F**) Cell density after 24 h of growth, reflecting proliferation rates of the cells. * *p* < 0.01, n.s.—not significant.

**Figure 6 biomedicines-13-02527-f006:**
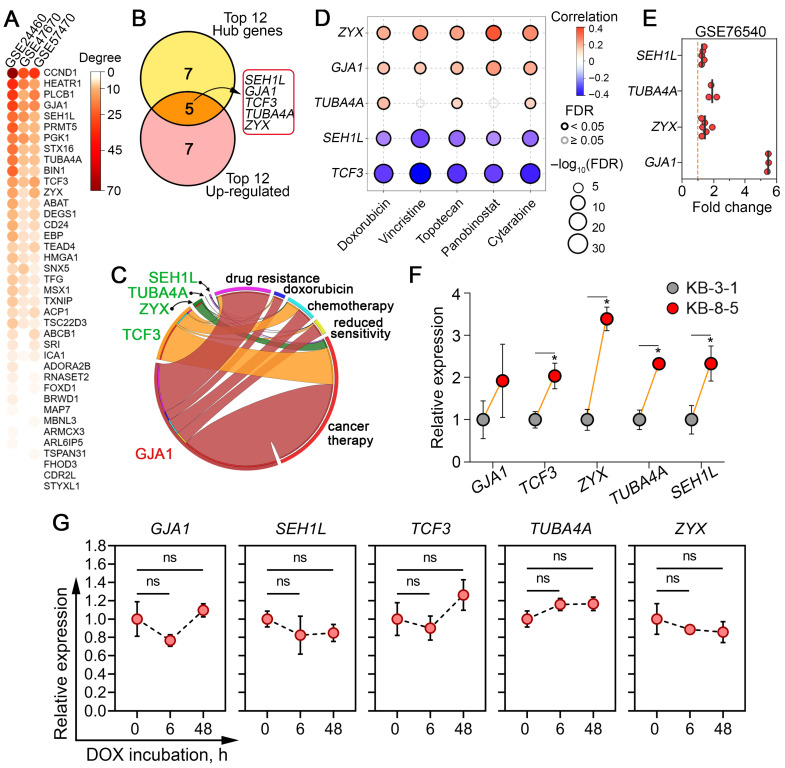
Selection of core genes demonstrating nodal positions within DOX resistance regulomes and verification of their activation. (**A**) A heatmap showing the number of neighbors (degree) of core genes within the gene networks reconstructed in Figure 3A. (**B**) Identification of hub genes with the greatest potential as marker candidates for DOX desensitization. The Venn diagram overlaps the top 12 hub genes with the highest degree scores with the top 12 the most up-regulated DEGs. (**C**) Text mining analysis revealed the co-occurrence of core genes with keywords associated with multidrug resistance in the abstracts of biomedical reports in the MEDLINE database using the GenClip3 tool. (**D**) Spearman correlation analysis between the upregulation of core genes and the overall sensitivity of a large panel of cancer cells to commonly used chemotherapeutic drugs, according to the Cancer Therapeutics Response Portal database. Blue and red bubbles represent negative and positive correlations, respectively; the deeper the color, the higher the correlation. The bubble size positively correlates with the FDR significance. The black outline indicates an FDR  <  0.05. (**E**) Expression levels of core genes in an independent cDNA microarray dataset (GSE76540) comparing DOX-resistant and DOX-sensitive breast cancer cells. (**F**) Expression levels of core genes in human KB-3-1 and KB-8-5 cervical carcinoma cells, as measured by qRT-PCR. *GAPDH* was used as a reference gene. (**G**) Expression of core genes in KB-3-1 cells after DOX (1.5 µM) exposure for 6 and 48 h assessed by qRT-PCR. *GAPDH* was used as a reference gene. Data are presented as mean ± SD. * *p*  <  0.05; ns, not significant.

**Figure 7 biomedicines-13-02527-f007:**
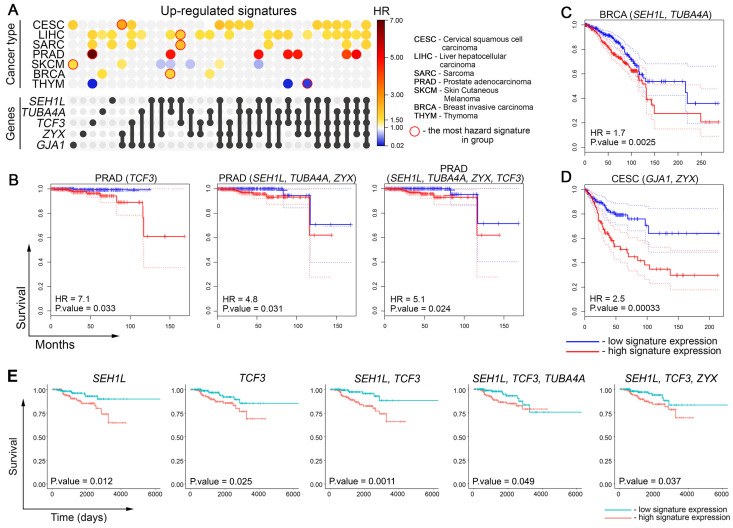
Relationship between activation of DOX resistance-related core genes with overall survival of cancer patients. (**A**) UpSet plot showing the hazard ratio (HR) of gene signatures containing the identified core genes in patients with different tumor types from The Cancer Genome Atlas (TCGA) database. Gene combinations are represented as sets in the UpSet panel, where filled circles and connecting lines indicate the genes included in each signature. The corresponding HR values for these combinations are displayed in heatmap format across cancer types. Only tumor types with statistically significant associations between gene expression and patient survival are presented. The most hazard signature for each cancer type is circled in red. (**B**–**D**) The Kaplan–Meier survival analysis of overall survival in patients from TCGA was stratified by the expression of the identified core genes. Patients were divided into high- and low-expression groups based on the median expression level of the gene signature. Survival curves are shown for (**B**) PRAD, (**C**) BRCA, and (**D**) CESC. Dashed lines indicate 95% confidence interval. (**E**) The Kaplan–Meier survival analysis of overall survival in BRCA patients treated with DOX, doxorubicin hydrochloride, and their liposomal forms (TCGA data). Patients were divided into high- and low-expression groups based on the median expression level of the gene signature.

**Figure 8 biomedicines-13-02527-f008:**
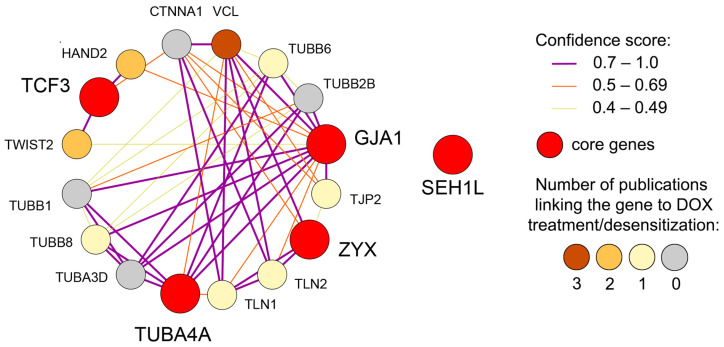
Gene network illustrating the interconnections among the identified core genes. The network, reconstructed from the STRING database, depicts both direct associations and those mediated by first-order gene neighbors. Gene neighbor nodes are colored according to the number of publications containing the gene name and the terms “drug resistance,” “doxorubicin,” “chemotherapy,” “reduced sensitivity,” or “cancer therapy” within a single sentence, as determined by Genclip3 text mining analysis.

**Table 1 biomedicines-13-02527-t001:** The GEO datasets used in the study.

GEO ID	Cells Origin	Resistant Cell Line	Parental Cell Line	Samples Number	Reference
GSE24460	Breast carcinoma	MCF-7ADR	MCF-7	4	[38]
GSE47670	Neuroblastoma	SK-N-SH-DR	SK-N-SH	6	[39]
GSE57470	Myelogenous leukemia	FEPS	K562	4	[40]
GSE76540	Breast carcinoma	MCF-7ADR	MCF-7	6	[41]

**Table 2 biomedicines-13-02527-t002:** Primers used in the study.

Gene	Type	Sequence
*GJA1*	Forward	5′-GATCGGGTTAAGGGAAAGAG-3′
	Reverse	5′-AGGAGACATAGGCGAGAG-3′
*SEH1L*	Forward	5′-ATAGCGACCAAAGATGTGAG-3′
	Reverse	5′-CGCCAGACCTGAGAATTATG-3′
*TCF3*	Forward	5′-AATAACTTCTCGTCCAGCC-3′
	Reverse	5′-GTGGTCTTCTATCTTACTCTGC-3′
*TUBA4A*	Forward	5′-ATCATTGACCCAGTGCTG-3′
	Reverse	5′-CTTGCCATAGTCAACAGAGAG-3′
*ZYX*	Forward	5′-GCCCTGGACAAGAACTTC-3′
	Reverse	5′-CATCTGCCTCAATCGACAG-3′
*GAPDH*	Forward	5′-ACCCCCAATGTGTCCGTCGT-3′
	Reverse	5′-TACTCCTTGGAGGCCATGTA-3′

## Data Availability

Upon reasonable request, the corresponding author will provide the data generated and/or analyzed during this study.

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
