# Peer review of "Novel Cross-Cancer Hub Genes in Doxorubicin Resistance Identified by Transcriptional Mapping"

_biomedicines, 2025, doi:10.3390/biomedicines13102527_

Round 1

Reviewer 1 Report

Comments and Suggestions for Authors The author of the paper titled "Beyond ABCB1: Pan-Cancer Hub Genes Governing Doxorubicin Resistance Revealed by Integrated Bioinformatic Analysis" has identified promising candidate biomarkers for universal DOX resistance through comprehensive bioinformatics analysis. The findings were further validated using cell experiments and data from the TCGA database. The author also acknowledged the limitations of the current study. Overall, this is a valuable and meaningful study that merits further in-depth investigation. However, I have several questions regarding specific aspects of the research and would appreciate it if the author could provide clarifications. p1. Has the cDNA microarray data from the Comprehensive Gene Expression Database (GEO) been previously reported in other studies? p2. The source of DOX used in the experiments was not specified. Please provide this information. p3. In the Doxorubicin accumulation assay, is the detection target Doxorubicin itself or the cells? Could the author explain the underlying principle of this assay? p4. What is the relationship between KB-3-1 and KB-8-5 cells? Are they representative of drug-resistant and non-drug-resistant phenotypes, respectively? Are there any other biological differences between these cell lines beyond drug resistance? p5. In Section 3.1, why is data from GSE76540 not included? Can this dataset be used to validate the results obtained from the other three datasets? p6. The study claims to identify cell context-independent processes involved in the acquisition of DOX resistance by cancer cells. Could the author elaborate on the analytical methods used to reach this conclusion? Specifically, how were irrelevant or context-specific changes accounted for in the analysis? p7. The author speculates that the overexpression of SEH1L and TCF3 represents a cellular compensatory response to exogenous stress. Could the author clarify which physiological functions are being compensated and what the biological purpose of this response might be? p8. In the discussion section, the author provides a detailed analysis of individual genes such as GJA1 and ZYX, which may dilute the focus on the core mechanisms related to DOX resistance. Did the author consider potential functional interactions among these genes and their collective role in DOX resistance?

Author Response

Dear Reviewer #1,

We would like to express our sincere gratitude for your thorough review and insightful comments on our manuscript. Your suggestions were instrumental in improving our study. In response to your comments, we have revised the manuscript and provide our detailed responses below. All changes in the text have been highlighted in yellow.

p1. Has the cDNA microarray data from the Comprehensive Gene Expression Database (GEO) been previously reported in other studies?

Authors: Corrected. Dear Reviewer #1, thank you for this important clarifying comment. The datasets we used (GSE24460, GSE57470, GSE47670, GSE76540) have indeed been previously reported in original studies, where they were deposited as part of the authors’ investigations of various cancer models and drug resistance. In our investigation we reanalyzed these data specifically to identify common differentially expressed genes (DEGs) associated with DOX resistance and to integrate the results across multiple tumor types using bioinformatic approaches. To the best of our knowledge, the combined comparative analysis of these datasets with a focus on cross-cancer mechanisms of drug resistance has not been previously reported. To clarify this point for readers, we have added links to original studies in Section 2.1.1 of Materials and Methods (please, see line 91 and Table 1) and Section 3.1 of Results (please, see lines 234, 236).

p2. The source of DOX used in the experiments was not specified. Please provide this information.

Authors: Addressed. We thank the reviewer for their careful reading of our manuscript. Indeed, we did not pay sufficient attention to specifying the source of DOX in the initial version of the manuscript. We have now corrected this and added the information of DOX reagent origin in the Section 2.2.2 of Materials and Methods (please, see line 167).

p3. In the Doxorubicin accumulation assay, is the detection target Doxorubicin itself or the cells? Could the author explain the underlying principle of this assay? 

Authors: Corrected. Indeed, the description of DOX accumulation assay may have seemed too brief. In this experiment, the detection target is DOX itself, as the compound has intrinsic fluorescence (excitation = 480 nm, emission = 590 nm). After incubation with DOX, the non-internalized drug was removed by washing cells with PBS. The intracellular accumulation of DOX was quantified by measuring the fluorescence intensity of the retained drug within the cells using flow cytometry. Thus, the principle of this assay is based on the fact that higher fluorescence intensity corresponds to increased intracellular DOX accumulation, which reflects the efflux activity of P-glycoprotein. To clarify this principle, we have added more detailed description of assay in Section 2.2.3 of Materials and Methods (please, see lines 173-180).

p4. What is the relationship between KB-3-1 and KB-8-5 cells? Are they representative of drug-resistant and non-drug-resistant phenotypes, respectively? Are there any other biological differences between these cell lines beyond drug resistance? 

Authors: Corrected. KB-3-1 cells are the parental human cervical carcinoma line, while KB-8-5 cells represent their multidrug-resistant derivative. KB-8-5 cells display stable overexpression of P-glycoprotein, which accounts for their low drug accumulation. Thus, KB-3-1 and KB-8-5 serve as well-established paired models of drug-sensitive and drug-resistant phenotypes, respectively. To answer your question and assess other biological differences in addition to the drug resistance phenotype, adherent properties and cholesterol level, we performed additional in vitro assays (assessment of morphology and proliferation rate (Sections 2.2.7 and 2.2.8 of Materials and Methods, please see lines 213-222)). It was shown that KB-8-5 cells were larger (Fig. 5A–D), displayed a more elongated, spindle-like morphology with an increased aspect ratio (major-to-minor axis length) (Fig. 5A, E), and demonstrated a lower proliferation rate (Fig. 5F). We have added this information to the revised manuscript in the Section 3.4 of Results (please, see lines 387-392 and Figure 5).

p5. In Section 3.1, why is data from GSE76540 not included? Can this dataset be used to validate the results obtained from the other three datasets? 

Authors: Addressed. We did not include the GSE76540 dataset in the initial comparative DEG analysis presented in Section 3.1 because it did not meet our dataset selection criteria (were added in revised manuscript in Section 2.2.1, lines 92–97). Specifically, both GSE76540 and GSE24460 were derived from the same breast cancer MCF7 cell line, whereas our cross-cancer analysis required datasets from different tissue origins. Nevertheless, GSE76540 was subsequently used as an independent dataset to verify the core genes identified in our study (Section 3.6), since it had been generated and published by other authors. Importantly, when the DEGs obtained from GSE76540 were compared with those from three other datasets (Figure R1), we observed that nearly all of the genes common to the three initial datasets were also presented in GSE76540 (13/16 of up-regulated and 18/21 of down-regulated). As expected, the strongest overlap was seen with GSE24460, which originated from the same MCF7 breast cancer cell model.

Figure R1. Venn diagrams showing the overlap of DEGs that were either upregulated or downregulated in each analyzed dataset. Genes common for three initially analyzed datasets are circled with red line.

p6. The study claims to identify cell context-independent processes involved in the acquisition of DOX resistance by cancer cells. Could the author elaborate on the analytical methods used to reach this conclusion? Specifically, how were irrelevant or context-specific changes accounted for in the analysis? 

Authors: Corrected. We thank the reviewer for raising this important point. To ensure that the identified processes truly reflect cell context–independent features of DOX resistance, we applied an intersection-based approach. Functional enrichment was first performed separately for each dataset, and only those GO terms consistently enriched across all three cancer cell types were selected. Terms unique to a single dataset were considered context-specific and excluded from downstream analysis. This filtering step allowed us to focus exclusively on processes reproducibly associated with DOX resistance across different tissue origins. We have revised the Section 2.1.2 of Materials and Methods (please, see lines 109–116) and Section 3.1 of Results (please, see lines 275–279) sections to clarify this procedure. We believe these edits have significantly improved the clarity of our methodology..

p7. The author speculates that the overexpression of SEH1L and TCF3 represents a cellular compensatory response to exogenous stress. Could the author clarify which physiological functions are being compensated and what the biological purpose of this response might be? 

Authors: Corrected. As previously mentioned in the Discussion part, SEH1L, as part of the NUP107–160 nuclear pore complex, maintains nucleocytoplasmic transport and ensures proper mitotic progression. Consequently, its overexpression may serve to compensate for stress-induced perturbations in nuclear transport and cell cycle regulation, thereby promoting proliferation under xenobiotic pressure. Indeed, several studies have shown the impact of nucleocytoplasmic transport into xenobiotic stress response [1–3]. In the case of TCF3, it has been demonstrated to regulate Wnt/β-catenin signaling , proliferation and EMT promotion [4,5]. We hypothesize that the upregulation of TCF3 can abrogate chemotherapy-induced antiproliferative signals, thereby maintaining stem-like properties and cellular plasticity. These may potentially enhance the survival of tumor cells during drug exposure. So, while alterations in SEH1L and TCF3 may not directly induce drug resistance, they can represent adaptive strategies to maintain survival pathways under conditions of cytotoxic stress. To clarify these suggestions for readers we revised Section 3.6 of Results (please, see lines 454-457).

p8. In the discussion section, the author provides a detailed analysis of individual genes such as GJA1 and ZYX, which may dilute the focus on the core mechanisms related to DOX resistance. Did the author consider potential functional interactions among these genes and their collective role in DOX resistance?

Authors: Corrected. We agree that the resistant phenotype is unlikely to be explained by single genes acting in isolation. In our revised Discussion (please, see lines 698-710), we now emphasize the functional convergence of the identified hub genes. Specifically, the gene network reconstruction using the STRING database (please, see Figure 8) revealed that four out of the five core genes are interconnected. These interactions occur either directly, such as between GJA1 and TUBA4A, or indirectly through first-order gene neighbors, some of which have documented links to DOX treatment or desensitization. Notably, ZYX, GJA1, and TUBA4A, involved in cytoskeleton and adhesion regulation, form stronger interactions with each other compared to TCF3, which regulates EMT-related transcriptional responses. SEH1L, associated with the nuclear pore, remains functionally separate. These findings suggest that, beyond individual gene effects, there is a partially coordinated transcriptomic response among core genes that may collectively contribute to tumor cell adaptation and resistance to prolonged DOX exposure. This new discussion block can be found on pages 20-21, lines 698-717.

We thank Reviewer #1 for their constructive criticism and highly valuable comments. Addressing these points has significantly strengthened our manuscript. We hope that corrected version of the manuscript will be acceptable for publication in the Biomedicines.

Sincerely,

Dr. Andrey Markov

References

  1. Marullo, R.; Rutherford, S.C.; Revuelta, M. V; Zamponi, N.; Culjkovic-Kraljacic, B.; Kotlov, N.; Di Siervi, N.; Lara-Garcia, J.; Allan, J.N.; Ruan, J.; et al. XPO1 Enables Adaptive Regulation of mRNA Export Required for Genotoxic Stress Tolerance in Cancer Cells. Cancer Res. 2024, 84, 101–117, doi:10.1158/0008-5472.CAN-23-1992.
  2. Turner, J.G.; Dawson, J.; Sullivan, D.M. Nuclear export of proteins and drug resistance in cancer. Biochem. Pharmacol. 2012, 83, 1021–1032, doi:https://doi.org/10.1016/j.bcp.2011.12.016.
  3. El-Tanani, M.; Dakir, E.-H.; Raynor, B.; Morgan, R. Mechanisms of Nuclear Export in Cancer and Resistance to Chemotherapy. Cancers (Basel). 2016, 8.
  4. Pu, X.-Y.; Zheng, D.-F.; Lv, T.; Zhou, Y.-J.; Yang, J.-Y.; Jiang, L. Overexpression of transcription factor 3 drives hepatocarcinoma development by enhancing cell proliferation via activating Wnt signaling pathway. Hepatobiliary Pancreat. Dis. Int. 2022, 21, 378–386, doi:https://doi.org/10.1016/j.hbpd.2022.01.003.
  5. Slyper, M.; Shahar, A.; Bar-Ziv, A.; Granit, R.Z.; Hamburger, T.; Maly, B.; Peretz, T.; Ben-Porath, I. Control of breast cancer growth and initiation by the stem cell-associated transcription factor TCF3. Cancer Res. 2012, 72, doi:10.1158/0008-5472.CAN-12-0119.

Reviewer 2 Report

Comments and Suggestions for Authors

Dear Editor,

The manuscript “Beyond ABCB1: Pan-Cancer Hub Genes Governing Doxorubicin Resistance Revealed by Integrated Bioinformatic Analysis” by Arseny D. Moralev et al., is reporting a valuable research on discovering hub genes involved in doxorubicin resistance. I enjoyed reading the manuscript in spite of several drawbacks, some of which have been also discussed by the authors. The research and the manuscript has been well organized, the data presentation and discussion are also very good. I would suggest some points that may help the authors to improve their paper.  

  • The title starts with ABCB1, while this term is rarely seen over the manuscript. Although it is attractive, considering that several other proteins are also involved in drug resistance, I would suggest to make some rewords to a shorter and stronger title.
  • There are too many abbreviations through the manuscript. Maybe it is better to change some of them to the full names.
  • Some parts of the introduction, lines 73-88, are actually parts of results and discussion.
  • The authors did not explain which criteria have been applied to select cDNA microarray profiles associated with DOX resistance from GEO database. More profiles are available there, using a quick search with the keywords.
  • The way the genes or their biological activities were selected or identified have not been well explained in some parts of the materials and methods; lines 103, 249, … .
  • Several of the figure legends are too short. It is better to include more details in the text or extend the relevant parts of the figure legends. For example, Figure 3C (similarity of clusters in what?) or Figure 3E, 6A, etc.
  • Line 37; double strand DNA break must change to “double strand DNA repair” or “DNA damage response”.
  • Line 479; needs to include some relevant references.
  • Line 578, TCF3 is “E protein (class I) family of helix-loop-helix transcription factors
  • A few cases need corrections for typo or grammar.

Author Response

Dear Reviewer #2,

We sincerely thank you for your thoughtful and positive assessment of our work, as well as for your valuable comments and suggestions. We greatly appreciate your insightful feedback.

We have edited the manuscript accordingly and have provided our responses to your points below. All changes made to the text have been highlighted in yellow for your convenience.

We hope that our findings will prove useful to other researchers in the field of chemoresistance and cancer treatment.

The title starts with ABCB1, while this term is rarely seen over the manuscript. Although it is attractive, considering that several other proteins are also involved in drug resistance, I would suggest to make some rewords to a shorter and stronger title.

Authors: Corrected. We appreciate the reviewer’s suggestion regarding the title. In response, we have revised it to “Novel Cross-Cancer Hub Genes in Doxorubicin Resistance Identified by Transcriptional Mapping”. This new title is shorter, stronger and better reflects the focus of our study on multiple core genes and their network-level contributions to DOX resistance across cancer types, rather than highlighting a single protein such as ABCB1.

There are too many abbreviations through the manuscript. Maybe it is better to change some of them to the full names.

Authors: Corrected. We agree with the reviewer's assessment that the manuscript was overly abundant with abbreviations. While common in bioinformatics literature, we recognize that excessive use can hinder readability. Following the reviewer's recommendation, we have carefully reviewed the manuscript and replaced several rarely used abbreviations with their full names where appropriate to improve readability and clarity. Specifically, we have removed the following abbreviations: GSCA (p. 4, line 134); CTRP (p. 4, line 135; p. 15, line 447; p. 17, line 526); NES (p. 8, line 304); FBS (p. 4, line 166); KEGG (p. 11, lines 366-367); FDPS and OSC (p. 18, line 591); and ROS (p. 2, line 47; p. 19, line 640).

Some parts of the introduction, lines 73-88, are actually parts of results and discussion.

Authors: Corrected. We thank the reviewer for this important comment regarding the manuscript's structure. We agree that lines 73–88 of the introduction included content more appropriate for the Results and Discussion sections. We have revised the manuscript to ensure that the Introduction now focuses solely on background, relevance, and study objectives, while the Results and Discussion present the findings and their interpretation (please, see lines 77-87).

The authors did not explain which criteria have been applied to select cDNA microarray profiles associated with DOX resistance from GEO database. More profiles are available there, using a quick search with the keywords.

Authors: Addressed. Indeed, there are more available datasets representing DOX-resistant tumor cells. To clarify the selection method, we chose cDNA microarray datasets from the GEO database according to the following criteria: (i) availability of both doxorubicin-resistant and corresponding doxorubicin-sensitive (parental) control cells, (ii) at least two samples per group, (iii) datasets derived from human cancer cell lines, and (iv) inclusion of different cancer types. Based on these criteria, three datasets were selected and are listed in Table 1. We have revised the Section 2.1.1 of Materials and Methods to state these inclusion criteria (please, see lines 93-97) to enhance clarity for the reader.

The way the genes or their biological activities were selected or identified have not been well explained in some parts of the materials and methods; lines 103, 249, … .

Authors: Corrected. Indeed, in some parts of Materials and Methods and Results the description of biological activities selection was too brief. We revised manuscript and added the information of how processes were identified and analyzed in Section 2.1.2 (please, see lines 109-116) and Section 3.1 (please, see lines 275-279).

Several of the figure legends are too short. It is better to include more details in the text or extend the relevant parts of the figure legends. For example, Figure 3C (similarity of clusters in what?) or Figure 3E, 6A, etc.

Authors: Corrected. Following this valuable recommendation, we have carefully revised the figure legends throughout the manuscript to provide more detailed descriptions, including the specific context of each panel. Please, see 3C, 3E, 6D, 7A-D. These changes indeed improve the clarity and self-sufficiency of the figures. Thank you!

Line 37; double strand DNA break must change to “double strand DNA repair” or “DNA damage response”.

Authors: Corrected (please, see lines 45-46).

Line 479; needs to include some relevant references.

Authors: Relevant references corresponding to the impact of cell respiration, metabolism, adhesion and viability on the adaptive response to cell stress were added (please, see line 556).

Line 578, TCF3 is “E protein (class I) family of helix-loop-helix transcription factors”

Authors: Corrected (please, see line 654).

A few cases need corrections for typo or grammar.

Authors: Corrected. Following the reviewer's comment, we have carefully re-read the entire manuscript. We identified and corrected several instances of awkward phrasing, typos, and redundant information. The text has been revised throughout; please see, for example, pages 8 (lines 298-301), 10 (lines 339, 341), 12 (lines 378-383), 13 (lines 410-411), and 15 (lines 457, 459, 470-473).

We thank Reviewer #2 once again for the time and effort dedicated to our manuscript. Their constructive criticism and insightful suggestions have been invaluable, significantly strengthening the clarity and impact of our work.

We hope the revised manuscript now meets the high standards for publication in the Biomedicines.

Respectfully,

Dr. Andrey Markov

Novosibirsk Academgorodok, Siberia, Russia

Reviewer 3 Report

Comments and Suggestions for Authors

In this article, the authors presents an integrated bioinformatics analysis of doxorubicin (DOX) resistance across three cancer cell types: leukemia, neuroblastoma, and breast cancer. Through network analysis and functional annotation, they identified five hub genes emerging as central regulators. The study includes experimental validation demonstrates that these hub genes correlate with poor clinical outcomes across multiple cancer types.

  1. Although the authors have included three different cancer types as well as experimental validations, I still think the dataset scope is limited, which weakens the generalizability of findings. I would suggest either doing a true pan-cancer analysis, or softening the “pan-cancer” claim.
  2. The survival analysis using TCGA data provides valuable clinical context, but the correlation between hub gene expression and overall survival doesn't necessarily translate to DOX-specific treatment outcomes. Validation in patient cohorts with known DOX treatment history would be more informative.
  3. The study doesn't address whether the identified signatures represent early adaptive responses or late-stage resistance mechanisms. Understanding the temporal dynamics would provide important insights for therapeutic intervention strategies.

Author Response

Dear Reviewer #3,

On behalf of all the authors, I would like to express our sincere gratitude for your thorough assessment of our manuscript, your constructive criticism, your positive view of our research, and your commitment to improving our article. We have revised the manuscript according to your valuable comments and provide our point-by-point responses below.

  1. Although the authors have included three different cancer types as well as experimental validations, I still think the dataset scope is limited, which weakens the generalizability of findings. I would suggest either doing a true pan-cancer analysis, or softening the “pan-cancer” claim.

Authors: Corrected. We fully agree that the number of analyzed transcriptomic datasets may not comprehensively represent the full diversity of tumor types. Our goal in this study was not to perform a complete pan-cancer analysis, but rather to identify cross-cancer transcriptional features that are shared among distinct cancer types exhibiting DOX resistance. To clarify this point, we have revised the manuscript to avoid any overstatement of generalizability. Specifically, we have replaced the terms “pan-cancer” or “universal markers” with more accurate expressions such as “cross-cancer,” “multi-cancer”. These changes have been implemented in the Title, Introduction (please, see line 81), Results (title of Section 3.3, Section 3.6 (line 473)), and Limitations (line 720) sections to better reflect the scope of our study.

  1. The survival analysis using TCGA data provides valuable clinical context, but the correlation between hub gene expression and overall survival doesn't necessarily translate to DOX-specific treatment outcomes. Validation in patient cohorts with known DOX treatment history would be more informative.

Authors: Corrected. We agree that overall survival data from TCGA may not directly reflect patient response to specific chemotherapeutic agents, including DOX, since treatment information in this database is often incomplete. To address this limitation, we have extended our analysis by focusing on the TCGA breast cancer (BRCA) cohort, which contains a subset of 363 patients with annotated DOX, DOX hydrochloride, or liposomal DOX treatment. The results, now presented in Figure 7E, demonstrate that higher expression of gene signatures that include SEH1L and TCF3 correlates with poor survival specifically in DOX-treated BRCA patients. These findings support the potential clinical relevance of the identified hub genes. Nevertheless, we acknowledge that validation in larger, clinically annotated patient cohorts with detailed DOX treatment histories will be necessary to further validate the predictive value of these genes (please, see Section 3.7 lines 513-524).

  1. The study doesn't address whether the identified signatures represent early adaptive responses or late-stage resistance mechanisms. Understanding the temporal dynamics would provide important insights for therapeutic intervention strategies.

Authors: Corrected. Dear Reviewer #3, we thank you for this insightful question and appreciate the valuable external perspective. We acknowledge that this distinction was overlooked in the original version of the manuscript. Your comment rightly highlights the importance of differentiating between early adaptive responses and stable, late-stage resistance mechanisms. To address this point, we have expanded our analysis by evaluating the expression of the identified hub genes in drug-sensitive KB-3-1 cells after short-term (6 h) and prolonged (48 h) exposure to a semi-toxic concentration of DOX (1.5 μM). The results, presented in Figure 6G, show that the expression levels of these genes were not affected at both time points, indicating that their upregulation in resistant KB-8-5 cells reflects established, late-stage resistance mechanisms rather than adaptive responses to DOX exposure. We have clarified this finding in the revised Section 3.6 of Results (please, see lines 463-468).

We believe the corrections and clarifications we have made have addressed all the reviewers' comments, and we hope the manuscript now meets the journal's standards for publication.

Thank you very much!

Sincerely,

Dr. Andrey Markov

Round 2

Reviewer 3 Report

Comments and Suggestions for Authors

In this revision, the authors have addressed previously raised issues. I have no further concerns.